# Exploring Food Addiction Across Several Behavioral Addictions: Analysis of Clinical Relevance

**DOI:** 10.3390/nu17071279

**Published:** 2025-04-06

**Authors:** Anahí Gaspar-Pérez, Roser Granero, Fernando Fernández-Aranda, Magda Rosinska, Cristina Artero, Silvia Ruiz-Torras, Ashley N Gearhardt, Zsolt Demetrovics, Joan Guàrdia-Olmos, Susana Jiménez-Murcia

**Affiliations:** 1Doctoral Program in Clinical and Health Psychology, University of Barcelona, 08007 Barcelona, Spain; agaspar@idibell.cat (A.G.-P.); silvia.ruiz@ub.edu (S.R.-T.); 2Department of Clinical Psychology, University Hospital of Bellvitge, 08908 Barcelona, Spain; ffernandez@bellvitgehospital.cat (F.F.-A.); mrosinska@idibell.cat (M.R.); cartero@idibell.cat (C.A.); 3Psychoneurobiology of Eating and Addictive Behaviors Group, Neuroscience Program, Bellvitge Biomedical Research Institute (IDIBELL), 08908 Barcelona, Spain; roser.granero@uab.cat; 4Ciber Fisiopatología Obesidad y Nutrición (CIBERobn), Instituto de Salud Carlos III, 28029 Madrid, Spain; 5Department of Psychobiology and Methodology, Autonomous University of Barcelona, 08193 Barcelona, Spain; 6Department of Clinical Sciences, School of Medicine and Health Sciences, University of Barcelona, 08007 Barcelona, Spain; 7Centre for Psychological Services, University of Barcelona (UB), 08035 Barcelona, Spain; 8Department of Psychology, University of Michigan, Ann Arbor, MI 48109, USA; gearhar@umich.edu; 9Institute for Mental Health and Wellbeing, College of Education, Psychology and Social Work, Flinders University, Adelaide, SA 5042, Australia; zsolt.demetrovics@gmail.com; 10Institute of Psychology, ELTE Eötvös Loránd University, 1053 Budapest, Hungary; 11Center of Excellence in Responsible Gaming, University of Gibraltar, Gibraltar GX11 1AA, Gibraltar; 12Facultat de Psicologia, Secció de Psicologia Quantitativa, Universitat de Barcelona, 08007 Barcelona, Spain; jguardia@ub.edu; 13UB Institute of Complex Systems, Universitat de Barcelona, 08007 Barcelona, Spain; 14Institute of Neuroscience, Universitat de Barcelona, 08007 Barcelona, Spain

**Keywords:** food addiction, addictive behaviors, clinical profile, gaming disorder, compulsive buying-shopping disorder, compulsive sexual behavior disorder

## Abstract

Background/Objectives: Recently, interest in studying food addiction (FA) in the context of behavioral addictions (BAs) has increased. However, research remains limited to determine the FA prevalence among various BAs. The current study aimed to investigate FA in a clinical sample of patients seeking treatment for gaming disorder, compulsive buying-shopping disorder (CBSD), compulsive sexual behavior disorder, and the comorbid presence of multiple BAs, as well as to determine the sociodemographic characteristics, personality traits, and general psychopathology of this clinical population. In addition, we analyzed whether FA is linked to a higher mean body mass index (BMI). Methods: The sample included 209 patients (135 men and 74 women) attending a specialized behavioral addiction unit. The assessment included a semi-structured clinical interview for the diagnosis of the abovementioned BAs, in addition to self-reported psychometric assessments for FA (using the Yale Food Addiction Scale 2. 0, YFAS-2), CBSD (using the Pathological Buying Screener, PBS), general psychopathology (using the Symptom Checklist-Revised, SCL-90-R), personality traits (using the Temperament and Character Inventory-Revised, TCI-R), emotional regulation (using Difficulties in Emotion Regulation Strategies, DERS), and impulsivity (using Impulsive Behavior Scale, UPPS-P). The comparison between the groups for the clinical profile was performed using logistic regression (categorical variables) and analysis of covariance (ANCOVA), adjusted based on the patients’ gender. The sociodemographic profile was based on chi-square tests for categorical variables and analysis of variance (ANOVA) for quantitative measures. Results: The prevalence of FA in the total sample was 22.49%. The highest prevalence of FA was observed in CBSD (31.3%), followed by gaming disorder (24.7%), and the comorbid presence of multiple BAs (14.3%). No group differences (FA+/−) were found in relation to sociodemographic variables, but the comorbidity between FA and any BA was associated more with females as well as having greater general psychopathology, greater emotional dysregulation, higher levels of impulsivity, and a higher mean BMI. Conclusions: The comorbidity between FA and BA is high compared to previous studies (22.49%), and it is also associated with greater severity and dysfunctionality. Emotional distress levels were high, which suggests that the group with this comorbidity may be employing FA behaviors to cope with psychological distress. However, a better understanding of the latent mechanisms that contribute to the progression of this multifaceted comorbid clinical disorder is needed. One aspect that future studies could consider is to explore the existence of FA symptoms early and routinely in patients with BAs.

## 1. Introduction

Studies in mental disorders have changed in recent years from a categorical to a dimensional diagnostic approach [1]. This dimensional concept can group a series of overlapping and interrelated symptoms, where different disorders may have elements on the same spectrum [2]. Impulsive-compulsive spectrum disorders (ICSDs) have dimensional and cross-dimensional symptoms that transcend traditional diagnostic boundaries and stem from this dimensional perspective [3]. These disorders share common characteristics, not only at the symptomatic level, but also at the level of etiology, comorbidity, trajectory of the disorder, and treatment outcome [2]. It has been postulated that these disorders are located along a continuum, with impulsivity at one extreme and compulsivity at the other. In the middle of this spectrum, both constructs can occur together, thus influencing each other’s development [3,4,5]. A common characteristic of all of them is the difficulty in evaluating the negative consequences of the behavior, giving priority to gratification, immediate pleasure, and/or short-term activation [6,7]. Recently, it has also been suggested that a transdiagnostic approach could provide insight into the difficulties in mental disorders; this transdiagnostic perspective focuses on the cognitive and behavioral processes common to different mental disorders that present comorbidly [1,8,9]. Moreover, in line with this, a factor that is important to both food addiction (FA) and behavioral addictions (Bas) is that of substance use [10,11].

Studies have found that substance and non-substance-related disorders often coexist, with a comorbidity rate ranging from 57.5% to 76.3% [12,13,14,15]. Various studies have demonstrated that there are common components in said disorders [16,17,18], ranging from personality traits and cognitive profiles [10,19,20] to genetic vulnerabilities, neurobiological mechanisms, and response to treatment [11,21,22,23,24,25].

Over the last decade, FA has attracted increasing scientific interest [26]. This condition describes specific maladaptive eating behavior patterns that are characterized by excessive consumption of ultra-processed foods with extreme food cravings as well as a loss of control [27,28,29]. Different studies and systematic reviews support the idea that FA can generate addictive-type behaviors [30] that are comparable to those seen in substance use disorder (SUD) or BAs [26,31,32,33] and that both share certain similar neural processes [32,34,35]. These results have also been supported by meta-analyses with neuroimaging studies that have reported that structures related to behaviors involving appetite and reward, notably the insula, striatum, amygdala, and orbital frontal cortex, which have a tendency to be triggered by both visual food and smoking cues [36]. Systematic reviews have also found that FA and SUDs share related pathways in the dopaminergic, opioid, and cannabinoid systems; dopamine has been associated with these clinical diagnoses as a reward mechanism that increases its release as a food or drug becomes more rewarding [35]. Regarding BAs, another meta-analysis based on neuroimaging studies identified that they show modified risk-related neural processes involving hyperactivity of the orbitofrontal cortex and striatum, as well as functional and structural damage in brain regions related to reward, decision making, and emotional processes [37,38].

In line with this, Jiménez-Murcia and colleagues [18] detected a 9.2% FA prevalence in patients seeking treatment for gambling disorder, with women presenting the highest percentage (30.5%). It was also found that FA shares characteristics with gambling disorder [39] and other BAs, such as compulsive buying-shopping disorder (CBSD) [40] (e.g., difficulties in controlling behavior, impulsivity, emotional dysregulation, craving related behaviors). Another study in patients with gaming disorder reported a higher prevalence of FA compared to patients with online gambling disorder [41]. In the general student population, some studies have also identified a link between FA and problematic use of the internet (PUI), as well as addictive phone use [42,43,44] and CBSD [45].

Commonly, in the case of BAs, the difficulty in resisting an urge or desire to perform a behavior despite negative consequences [21,46,47,48] can also lead to significant impairment in social, familial, personal, occupational, educational, or other important life domains [49,50]. The co-occurrence of multiple addictive disorders, including FA, is often associated with greater symptomatology [18], more general psychopathology [39,51], and more dysfunctional personality traits [52], but also with a poorer treatment outcome [24]. However, there is a lack of studies in the literature, with the exception of gambling disorders, exploring the characteristics of patients with BA and FA and their interaction. The only studies that have investigated this relationship, specifically in gambling disorder, have found that being female and younger in age were associated with the presence of FA [18,39] and that those patients also exhibited more severe pathology. Only a previous study by Müller et al. [40] examined the relationship between FA and different BAs, although solely looking at a sample of bariatric surgery candidates. They found the FA group had higher psychopathology. Since these data are related to a specific group, further research is needed in relation to other groups and the general population.

There is previous evidence that FA presents comorbidities with other mental illnesses, the main ones being anxiety and mood disorders, as well as eating disorders (EDs) [53,54]. However, due to the limited research about FA in BAs and the impact they may have on the subjects’ life functioning, the present article’s principle aims are the following: (a) to identify the prevalence of FA in a clinical sample of patients pursuing treatment for gaming disorder, CBSD, compulsive sexual behavior disorder, and the comorbid presence of multiple BAs; and (b) determine the sociodemographic characteristics, personality traits, and general psychopathology of this clinical population. Additionally, a secondary aim of this study was to find whether FA is linked to a higher mean body mass index (BMI). To our knowledge, the studies which have looked at FA and BAs were in specific populations (e.g., bariatric patients). This is the first one to examine a clinical population, excluding gambling disorder, investigating FA in treatment-seeking individuals with gaming disorder, CBSD, and compulsive sexual behavior disorder as primary diagnoses and with multiple diagnoses. Hence, it aims to provide a different perspective on the information currently available. We hypothesize that participants with FA and these BAs will have a specific profile that distinguishes them from others who do not have FA. Also, we hypothesize that the presence of FA among gaming disorder, CBSD, compulsive sexual behavior disorder and the comorbid presence of multiple BAs could have an association with worse clinical profiles, greater psychopathology, and greater severity of the disorder.

## 2. Materials and Methods

### 2.1. Participants and Procedure

The sample included N = 209 patients (135 men and 74 women). They all received treatment from the Behavioral Addictions Unit of the Department of Clinical Psychology at Bellvitge University Hospital between June 2016 and December 2023. The inclusion criterion was that the patients requested treatment for gaming disorder, CBSD, or compulsive sexual behavior disorder. Neither the diagnostic criteria for any other addiction nor a current ED, according to the Diagnostic and Statistical Manual of Mental Disorders (DSM-5), were met by any of the participants. Because FA has been associated with EDs [54,55], the presence of any type of ED was considered an exclusion criterion to examine the unique characteristics of FA in BAs. Patients with incomplete data in the measurement tools were also excluded (33 participants from the 242 initial candidates). No form of compensation was provided to patients for participation in this study, and signed informed consent was received from each participant.

### 2.2. Assessment

#### 2.2.1. Semi-Structured Clinical Interview

The presence of a BA was evaluated using a semi-structured face-to-face clinical interview conducted by clinical psychologists who have over 20 years of experience in the diagnosis of BAs and EDs. In addition, other data that were collected were sociodemographic features, education level, employment status, marital status, and the socio-economic position index according to Hollingshead’s scale (this scale produced a classification subject to the participants’ education level, their employment status, and professional prestige) [56], among other relevant indexes. The BAs in this study were assessed using the following criteria: Gaming disorder [57,58] was assessed using DSM-5 criteria and validated within a 12-month period to establish the diagnosis [59,60]. Compulsive buying-shopping disorder (CBSD) was contingent on the subsequent guidelines that have been established by McElroy and colleagues (1994). Although the validity and reliability of these guidelines are yet to be determined, they have been widely acknowledged by the scientific community [61,62]. Regarding the assessment of compulsive sexual behavior disorder, a list of self-reported items was used. These items were based on the consensual definition in the DSM-IV-TR [63] in the Sexual Disorders Not Otherwise Specified section (302.9). The criteria that were used in this study to identify patients seeking treatment for compulsive sexual behavior were not based on the most recent versions of the DSM since, although this disorder was suggested to be included in the DSM-5, it was eventually dropped. In the proposal, criteria such as an exorbitant amount of time spent on sexual activity, using sex as a means to regulate negative affective states, impaired self-control, and the persistence of the behavior regardless of the negative consequences were incorporated [64]. However, it was considered that further research was needed based on community samples that had not been pre-screened for hypersexuality or other forms of psychopathology. Therefore, it was not included as a mental disorder in this manual, with the argument that this avoided pathologizing sexual activity, as there was insufficient scientific support. However, this disorder was included in the International Classification of Diseases, 11th ed. (ICD-11), as “Compulsive Sexual Behavior Disorder” and classified under Impulse Control Disorder (ICD). The items included in the screening used in the present study are completely aligned with those that were collected for the ICD-11 for this disorder. In addition, an experienced psychologist subsequently confirmed the diagnosis in a semi-structured, face-to-face clinical interview.

#### 2.2.2. Self-Report Measures

Each participant completed self-report questionnaires to analyze psychopathological symptoms, personality traits, sociodemographic and other pertinent clinical variables.

The Yale Food Addiction Scale 2.0 (YFAS-2) [65]: This self-report questionnaire has been adapted to determine addictive eating behaviors based on the DSM-5 criteria for SUD. In the current study, this questionnaire was used to measure FA presence. It assesses 11 symptoms in 35 items on an eight-point Likert-type scale ranging from 0 = never to 7 = every day. Severity limits are established as follows: mild (2 to 3 symptoms), moderate (4 to 5 symptoms), and severe (6 to 11 symptoms). The Spanish validation of the YFAS-2 [66] reported an internal consistency of 0.94 (alpha coefficient). In the present study, the internal consistency of the total score was α = 0.90 (see Appendix A). The positive screening threshold was based on the specific criterion defined in the psychometric validation studies and consisted of the presence of a minimum of two symptoms moreover the presence of clinically significant impairment or distress.

Pathological Buying Screener (PBS) [67]: This is a 13-item scale that was translated to Spanish (from the original English) by Fernández-Aranda et al. (2019) [68] following the International Test Commission Guidelines for Translating and Adapting Tests 2010. Retrieved from https://www.intestcom.org/files/guideline_test_adaptation.pdf (accessed on 31 March 2025). The internal consistency in this study was good (α = 0.86). Cronbach’s alpha for Time 1 was 0.85 and for Time 2 was 0.84.

Symptom Checklist-Revised (SCL-90-R) [69]: This questionnaire is used to measure various psychological and psychopathological symptoms and is composed of 90 items that measure nine dimensions of primary symptoms: somatization, obsession-compulsion, interpersonal sensitivity, depression, anxiety, hostility, phobic anxiety, paranoid ideation, and psychoticism. Additionally, this test yields the following: (a) a global severity index (GSI), (b) a positive symptom distress index (PSDI), and (c) a positive symptom total (PST). This instrument has been validated in the Spanish population [70] and has reported a good internal consistency ranging from 0.81 to 0.90, with re-test reliability ranging from 0.78 to 0.90. The internal consistency in our sample in the total score was α = 0.98 (see Appendix A).

Temperament and Character Inventory-Revised (TCI-R) [71]: This is a questionnaire with 240 items that assesses seven personality dimensions: four associated with temperament (novelty seeking, harm avoidance, reward dependence, and persistence) and three with character (self-direction, cooperation, and self-transcendence). The Spanish version [72] has been well-documented. The reliability of the seven dimensions in the Spanish adaptation range between 0.77 and 0.84.

Impulsive Behavior Scale (UPPS-P) [73]: This is a self-report questionnaire with 59 items that measures five facets of impulsive behavior: negative urgency (NU), lack of perseverance (LP), lack of premeditation (LPM), sensation seeking (SS), and positive urgency (PU). The Spanish version [74] of the UPPS-P was used for this study. For the total score of this scale, the internal consistency in our sample was α = 0.90 (see Appendix A).

Difficulties in Emotion Regulation Strategies (DERS) [75]: This assesses emotional dysregulation using a self-report with 36 items divided into six subscales: non-acceptance of emotional responses, difficulties engaging in goal-directed behavior when having strong emotions, impulse control difficulties, lack of emotional awareness, limited access to emotional regulation strategies, and lack of emotional clarity. Higher scores indicate greater problems with emotion regulation. This instrument has been validated in the Spanish population [76]. In this study, the internal consistency of the total score was α = 0.93 (see Appendix A).

### 2.3. Statistical Analysis

The data analysis was carried out with Stata18 for Windows [77]. The comparison of the sociodemographic profile between the two groups of the study (defined by the YFAS-2 screening result, negative versus positive) was based on chi-square tests for categorical variables and analysis of variance (ANOVA) for quantitative measures. The comparison between the groups for the clinical profile was done with logistic regression (categorical variables) and analysis of covariance (ANCOVA), adjusted based on the patients’ gender. The goodness of fit for the logistic regressions was tested with the Hosmer–Lemeshow test (adequate fitting was considered for *p* > 0.05). Regarding the ANOVA-ANCOVA, these procedures are strongly robust to potential violations of typical assumptions, such as normality and homoscedasticty, particularly with large datasets (samples sizes higher than 30 are typically recommended for reliable employment of the tests). The effect size of the proportion differences was estimated with the standardized coefficient Cramer-V (C-V, a mild-moderate to high-large effect size was considered for values above 0.20), and the effect size for the mean differences was calculated with partial eta-square (η_p_^2^, considering mild-moderate to high-large effects for values higher than 0.10) and Cohen’s-*d* (considering mild-moderate to high-large effects for values higher than |*d*| > 0.50) [78]. Finner’s correction (an alternative method to the classical Bonferroni’s correction) was used to avoid the increase in Type I error due the use of multiple statistical significance tests [79].

## 3. Results

### 3.1. Description of the Sample

Amid the total sample, the distribution of the patients’ gender was *n* = 74 women (35.4%) versus *n* = 135 men (64.6%). Most participants were single (*n* = 141, 67.5%), achieved a secondary education level (*n* = 87, 41.6%), were unemployed (*n* = 129, 61.7%), and were grouped into mean-low to low social position indexes (*n* = 148, 70.9%). The mean age was 35.5 years (SD = 15.5). The comparison between patients who achieved a negative FA screening versus a positive one only achieved differences in the gender distribution (higher proportion of women among the group of patients with a positive FA screening). Table 1 shows the detailed frequency distribution of all the sociodemographic variables in the study.

### 3.2. Presence of FA and Comparison Between Behavioral Addiction Subtypes

The number of patients who screened positive for FA in the whole sample was *n* = 47 (22.49%), and the mean severity of FA (as measured by the YFAS-2 total) was 2.50 (SD = 3.3). Estimates stratified by the groups defined by the BA subtype and the results of the comparison between the groups (adjusted for the patients’ gender) are displayed in Table 2 (see also Figure 1).

The highest prevalence for a positive FA screening was identified in patients with CBSD (31.3%), followed by gaming addiction (24.7%), the comorbid presence of multiple BAs (14.3%), and compulsive sexual behavior disorder (4.5%). Similar results were obtained considering the FA severity levels, with the highest means associated with CBSD (3.14) and the lowest with compulsive sexual behavior disorder (1.43).

### 3.3. Variables Associated with the Presence of a Positive FA Screening

The results of the ANCOVA (adjusted for sex) comparing the groups defined for patients with FA negative versus positive screening scores are displayed in Table 3. According to these results, the presence of a positive FA screening was associated with a higher mean BMI, higher likelihood of psychological distress (higher mean scores in all the SCL-90R scales), higher impulsivity (concretely in the UPPS-P positive urgency, negative urgency, and total scale), increased difficulties in emotion regulation strategies (except in the DERS lack of awareness scale), higher harm avoidance, and a lower mean in persistence and self-directedness.

Figure 2 displays the radar chart with the standardized means in the main clinical variables analyzed in the study, as a synthesis of the results of the comparison between the groups defined for patients with negative FA screening versus positive FA screening.

## 4. Discussion

In this study, we aimed to explore FA in patients with various BAs as a main disorder (namely gaming disorder, CBSD, compulsive sexual behavior disorder) and to compare sociodemographic characteristics, personality traits, and general psychopathology among the groups (with and without FA). The results indicated a prevalence of FA in the total sample of 22.49% (57.4% in women and 42.6% in men). The highest prevalence of FA was observed in CBSD (31.3%), followed by gaming disorder (24.7%), the comorbid presence of multiple BAs (14.3%), and compulsive sexual behavior disorder (4.5%). These results are higher than those reported in gambling disorder samples with FA (8.3%) [18,39] and those reported in non-clinical populations (between 3% and 20%) [54]. Müller et al. [40] reported a significant association between FA symptoms and CBSD and PUI symptoms in patients with obesity who were candidates for bariatric surgery. Another study found a higher prevalence of FA in patients with gaming disorder compared to patients with online gambling disorder [41]. The lowest prevalence of FA in the present study was found in patients with compulsive sexual behavior disorder (4.5%). Only in the general population has a relationship between FA and compulsive sexual behavior disorder been reported [45]. In this sense, the literature is scarce in determining the prevalence of FA other than gambling disorder. In our case, the prevalence of FA in patients who already had comorbidity with multiple BAs was 14.3%. As in the case of SUD [24], this result may suggest, as in other studies, that addictions often coexist [11,25].

We consider it important to note that previous research has described a high percentage (60%) of overlap between FA and EDs [54]. Although, the results of the present study show that FA is also prevalent at a high level in samples without EDs (22.49%), this could mean that FA is a clinically significant construct related to, but distinct from EDs. Although FA is a controversial condition, it is still possible to assess it using the current diagnostic manuals of mental disorders. The scale is based on the DSM-5 criteria for substance use disorders and was developed by Ashley Gearhardt [65]. Since its publication in 2009 [27], research interest in this condition has grown exponentially. Epidemiological studies show that between 2% and 12% of the general population has FA [80,81]. Although in diseases such as EDs, the prevalence of FA is particularly high. Therefore, some studies point to the existence of a collinearity between FA and ED symptoms (especially BED) [66]. Thus, it could be considered, albeit cautiously, that FA is a disorder per se, although it would show a high comorbidity with other specific conditions. However, all this does not contradict the usefulness of the dimensional classification of mental disorders, which would have some advantages over the categorical one. Precisely in aspects such as comorbidity. From this dimensional perspective, different spectrums could be identified, such as an impulsive–compulsive spectrum, which would include conditions that share common risk factors and similarities, but also differences [2]. From this dimensional approach, disorders such as addictive disorders (substance and behavioral), FA, ADHD (attention deficit hyperactivity disorder), EDs, obsessive-compulsive and body dysmorphic disorders could be considered [6]. Thus, FA can be seen as one condition, but highly prevalent in other disorders that share common risk factors [54]. To date, multiple studies have examined whether the presence of FA in EDs is related to greater severity as well as worse response to treatment [82,83]. However, few have been carried out in relation to other disorders, such as behavioral addictions. In fact, the exception has been in gambling disorder. Therefore, the present research aims to analyze the presence of FA in other BAs by clinically characterizing patients suffering from both conditions.

For BAs, some treatment approaches have been proposed however, gambling disorder is the only one that has valid and evidence-based treatments [84]. In the field of FA, few interventions are available, and their efficacy is unclear [35]. Currently, research has focused more on the conceptualization of both FA and BAs other than gambling disorder, which has limited the development of clinical guidelines for both FA and BAs, and such guidelines are now beginning to be created [85,86,87]. For FA, systematic reviews have reported that interventions focused on lifestyle modification (diet and physical activity), pharmacological interventions (combination of naltrexone and bupropion, as well as pexacerfont), and bariatric surgery report a decrease in FA symptoms [88]. Similarly, a meta-analysis has reported that psychological treatments with a cognitive-behavioral approach as well as pharmacological and mixed treatments decrease the overall severity of these BAs in the short term [23]. Other positions consider that the creation of specific protocols for each BA is a factor that limits treatment, not only because of the large number of protocols that would be necessary, but also because it involves training several professionals in this field as well as time and high costs for the institutions [84]. Treatments targeting transdiagnostic mechanisms are not new; they have already been applied in anxiety disorders, FA, and EDs and for BAs and SUD [17]. Because addictive disorders have similar underlying mechanisms and clinical features, some research is proposing transdiagnostic treatments for all of them [17]. Therefore, we support the position of creating such treatments to address FA + BA comorbidity.

In this study, greater FA severity was associated with CBSD. No group differences (FA+ vs. FA−) were found with respect to sociodemographic variables (age, marital status, educational level, or social status). In both groups, most participants were single, with a primary and secondary level of education, unemployed, and with a low social position index. The mean age was 35.46 (SD = 15.50), indicating a younger mean age than that reported in samples with gambling disorder (40.5 years) [41]. These findings show similarities with the sociodemographic profiles identified in other studies in samples with BAs [39,89,90,91]. However, age is a sociodemographic variable that could be considered further, as one study identified that being younger than 50 years of age is a sociodemographic variable associated with an increased risk of psychiatric comorbidity [92].

The comorbidity between FA and any BA was associated more with females. These results are similar to those reported in other studies of patients with gambling disorder that found that being female and younger were two factors associated with the coexistence of the two conditions [18,39]. Previously, the literature has identified that women are more likely to present with FA [81,93,94]. In relation to CBSD, it has been identified that the likelihood of developing such behavior is more frequent in women compared to men [95,96,97]. Evidence also indicates that people with a diagnosis of CBSD are almost twice as likely to have comorbidity with substance use, and three times as likely to develop an ED [98]. Regarding gaming disorder, which was the second most prevalent BA with FA in this study, in the literature, it has been documented that this disorder affects more men than women [99,100]. In terms of FA, however, a meta-analysis reported that there was a higher prevalence in women, which could be attributed to gender-related differences in hormonal profiles and/or dietary patterns [80]. In addition, women are more likely to engage in addictive behaviors to cope with negative emotional states (such as anxiety, stress, and depression) [101]. Although, because of the limited research regarding FA in all BAs, the impact that gender may have on the comorbidity between FA and this BA is not precisely known. Another important aspect to consider is that due to the cross-sectional nature of this study, we cannot determine the causality of these two variables, which justifies the development of future lines of research and longitudinal studies.

The presence of FA was also associated with greater global psychological distress and more difficulties in emotion regulation strategies (except for lack of conscientiousness). These outcomes are similar to those reported by Jiménez-Murcia et al. [18] who determined that the coexistence of FA and gambling disorder is related to poorer emotional and psychological states. Other studies have previously reported that BAs are associated with deficits in emotional regulation (ER) [102,103]. However, our findings suggest that these deficits are still greater in patients with FA comorbidity, and that ER should be considered an important factor in this comorbidity and could even be a target for treatment in these patients. It has also been suggested that some addictive behaviors, such as gambling disorder and FA, are dysfunctional strategies used to regulate negative mood states [18]. In addition, evidence indicates that psychiatric comorbidities may affect emotion regulation and that it is likely that emotional dysregulation may be a predictive factor in the development of addiction [102,104]. Although our results provide additional evidence of ER as a transdiagnostic factor in BAs, a better understanding of the relationship among FA, BAs, and ER is needed.

This study showed that individuals who present with a BA and FA have been seen to be significantly more impulsive compared to those without FA, in terms of the UPPS-P (higher positive and negative urgency in addition to total score). This could suggest that these individuals find an increased need to act impulsively regardless of whether they experience positive or negative emotions, without taking into account any possible longstanding effects. This finding also shows that the differences between the groups (FA+ vs. FA−) are observed in emotional impulsivity and not in cognitive impulsivity, and that emotional impulsivity is higher in patients with this comorbidity, having both a positive FA screening and a diagnosed BA (FA + BA). In line with this, a recent systematic review analyzing impulsivity in clinical samples with obesity and FA identified that cognitive impulsivity seems to be more related to the male sex [105]. This could be a possible explanation for the result we obtained; however, we cannot interpret it in this way until we have more scientific evidence to support this potential explanation. It is possible that both FA and BA would have the functionality of producing gratification (positive reinforcement) and alleviating discomfort (negative reinforcement). In this context, reinforcement theory establishes that decreasing negative affect and increasing positive reinforcement are the main reinforcers and motivators of maladaptive behaviors [106,107]. An individual may act rashly in the face of intense positive emotions and during intense negative emotions [84]. Delaying gratification is an aspect of self-regulation and self-control, but an individual can engage in addictive and food-related behaviors and ignore any negative effects by prioritizing immediate gratification [35]. Previous research has found that negative urgency is higher in individuals with gambling disorder and problematic pornography use compared to those with gambling disorder alone [108]. In contrast, positive urgency has been seen to be associated with FA [109]. Moreover, when looking at impulsivity and food addiction, which has mainly been observed in EDs [110,111], it has been seen to be a factor in FA development and maintenance [105].

Also, these individuals (FA + BA) presented higher harm avoidance and lower persistence and self-directedness. This could imply that they tend to be more anxious or fearful when going through life, as well as less persistent and/or indecisive about the situations they experience, and lastly less able to adapt to situations and reach their objectives or goals. Previous studies have shown that individuals with CBSD and compulsive sexual behavior disorder tend to have higher harm avoidance and lower self-directedness [112,113,114,115], and those with gaming disorder also have lower self-directedness, especially when related to the likelihood of suicidal behavior [116]. However, personality traits in relation to the comorbidity of these disorders (BA + FA) have not been explored. However, in patients with gambling disorder and high FA scores, an association has been found where they had higher harm avoidance [18]. These personality traits may have vast implications for the long-term treatment and potential prolongation of these co-concurrent addictions, therefore novel methods of addressing them are required.

Additionally, regarding the secondary objective of BMI, which was calculated based on self-reported weight and height, the presence of FA was associated with a higher mean BMI (28.51) compared to the group without FA (25.24). The World Health Organization (WHO) has established a set of criteria for classifying BMI as underweight or normal weight (BMI < 25), overweight (25 ≤ BMI < 30), and obese (BMI ≥ 30) [117]. According to these criteria, participants with FA + BA are overweight. Moreover, other studies in samples with gambling disorder and gaming disorder have reported an association with a high BMI (obesity) [39,41,118]. High BMI has also been linked to EDs and PUI [119]. Likewise, in relation to FA, studies have found a relationship with higher BMI [80,120]. However, FA can also occur in different weight categories, as some studies have also reported the presence of FA in samples with a BMI within the range of having anorexia nervosa [29,81]. These results increase the necessity to further explore the relationship between FA, higher BMI and different BAs. It is known that factors such as lifestyle (including exercise level, eating habits, tobacco and alcohol consumption, or sleep quality) as well as genetic or metabolic factors may have an influence on BMI [121,122,123]. However, these aspects were not examined in this study.

Finally, there is limited research on FA and BA from the clinical perspective, especially those other than gambling disorder. More studies are investigating how comorbidities may influence the maintenance, development, and treatment of ICSD, as scientific evidence has found an association in psychiatric comorbidities with increased risk of mortality, as well as increased symptom severity, multiple physical symptoms, stress, and poorer general health [92]. Although the exploration of the BA field has been scant, a recent meta-analysis identified an overall prevalence of 11.1% regardless of BA type [124].

## 5. Limitations and Strengths

In this study, there were some limitations that ought to be taken into account when interpreting the findings. First, the groups (FA+ vs. FA−) were uneven in terms of sex; therefore, future research should consider including more balanced samples. Second, each participant was recruited from a hospital setting; therefore, the results may not be representative of the general population. Third, the scales assessed were self-reported, which could be a limitation in itself. The information on weight and height used to obtain BMI was also self-reported, so this result might not be accurate, as they were not measured by the investigators. Fourth, the sample size for the group of patients meeting criteria for FA was small. This may have an effect on the statistical analyses performed, which may possibly be underpowered. Although, it is important to consider that there is a lower prevalence of other BAs in clinical practice when compared to gambling disorder. Furthermore, the cross-sectional nature of this study did not allow for the inclusion of predictive models (to identify the possible variables with significant influence on the presence and severity of FA), a mediational analysis aimed at identifying the pathways contributing to the comorbid presence of FA with BAs, or developmental trajectories of the course of this clinical concurrence.

Despite these limitations, the present study also presents several strengths. First, it is a novel study that considers FA in patients with BAs other than gambling disorder. Second, it compares several BAs that are not normally analyzed together. Since the sample consisted of patients treated in a specialized BA unit, the evaluations were performed in a uniform manner. The findings obtained in the present study demonstrate the need for further attention in the field of BAs.

## 6. Conclusions

The findings of this study demonstrate that comorbidity between FA and BAs is high, compared to previous studies (22.49%), and it is also associated with a clinical profile of greater severity and more dysfunctionality. These results also indicate that FA exerts a negative influence on BAs. The high levels of emotional distress suggest that the group with this comorbidity may be employing FA behaviors to manage psychological distress. Therefore, it is important to explore the existence of FA symptoms early and routinely in treatment-seeking BA patients. We believe that it would be worthwhile to incorporate their detection into the diagnostic protocols of all BAs, including those not addressed in this study using valid and reliable instruments, as well as experts in the field of addictions. In this clinical context, it is important to inquire about the clinical relevance of other mental health conditions that may be involved in the development, maintenance, and response to the treatment of BAs. In addition, more large-scale epidemiological studies in BAs other than gambling disorder and greater attention to the BA field in general are needed. Also, to obtain a better understanding of the underlying mechanisms that contribute to the course of this clinically complex comorbid condition, neurobiological research and high quality longitudinal studies would help to provide a more solid basis in this regard. Similarly, there is a need to increase knowledge about FA in the context of mental health conditions overall.

## 7. Clinical Implications

Little is known about BAs other than gambling disorder in general. A better understanding of the factors involved in the relationship of BAs with FA could have relevant implications for future treatment designs. Future clinical research could further investigate the role of FA in the treatment response to BAs. In this sense, considering aspects such as comorbidity and associated addictive components could help in the design of transdiagnostic treatments for addictions. According to the findings of this study, using a cognitive-behavioral approach, the focus was mainly on variables such as impulsivity; emotional dysregulation; cognitive aspects, such as decision making; and behavioral aspects, such as lifestyle. An important challenge in the clinical setting is the creation of high quality and validated treatment protocols, since to date, there is a clinical void in this regard.

## Figures and Tables

**Figure 1 nutrients-17-01279-f001:**
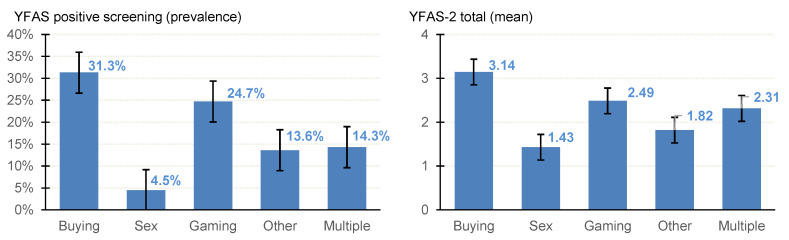
Prevalence of positive FA screening and mean of the YFAS-2 total. Note. Vertical lines represent error bars.

**Figure 2 nutrients-17-01279-f002:**
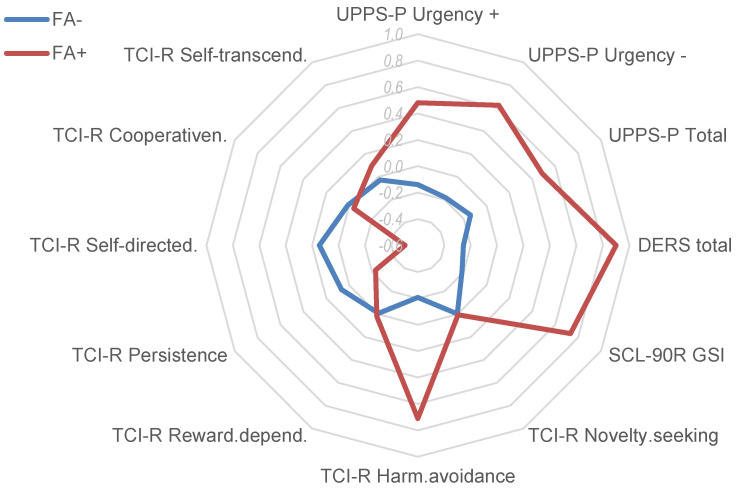
Radar chart. Note. FA−: negative food addiction screening. FA+: positive food addiction screening. Means for z-standardized scores are plotted.

**Table 1 nutrients-17-01279-t001:** Comparison of sociodemographic variables.

	Total		FA−		FA+			
	(*n* = 209)		(*n* = 162)		(*n* = 47)			
	*n*	%	*n*	%	*n*	%	*p*	C-V
Sex								
Female	74	35.4%	47	29.0%	27	57.4%	**<0.001 ***	**0.248 ^†^**
Male	135	64.6%	115	71.0%	20	42.6%		
Marital status								
Single	141	67.5%	109	67.3%	32	68.1%	0.972	0.017
Married—couple	53	25.4%	41	25.3%	12	25.5%		
Divorced—separated	15	7.2%	12	7.4%	3	6.4%		
Education								
Primary	78	37.3%	56	34.6%	22	46.8%	0.257	0.114
Secondary	87	41.6%	69	42.6%	18	38.3%		
University	44	21.1%	37	22.8%	7	14.9%		
Employment								
Unemployed	129	61.7%	95	58.6%	34	72.3%	0.089	0.118
Employed	80	38.3%	67	41.4%	13	27.7%		
Social								
High	11	5.3%	10	6.2%	1	2.1%	0.615	0.113
Mean-high to high	34	16.3%	27	16.7%	7	14.9%		
Mean	16	7.7%	14	8.6%	2	4.3%		
Mean-low	53	25.4%	40	24.7%	13	27.7%		
Low	95	45.5%	71	43.8%	24	51.1%		
	**Mean**	**SD**	**Mean**	**SD**	**Mean**	**SD**	** *p* **	** *|d|* **
Age (years)	35.46	15.50	35.41	16.06	35.62	13.56	0.937	0.01

Note. FA−: negative food addiction screening. FA+: positive food addiction screening. SD: standard deviation. * Bold: significant comparison (0.05 level). ^†^ Bold: relevant effect size (coefficient into the mild-moderate to large-high effect range).

**Table 2 nutrients-17-01279-t002:** Prevalence of positive FA screening and mean of the YFAS-2 total: results adjusted for sex.

	FA+ (Positive YFAS-2 Screening)				Pairwise Comparisons					
	*n*	Prev	95% CI		Contr.	^1^ *p*	^1^ OR	Contr.	^1^ *p*	^1^ OR
Buying (B)	21	31.3%	20.24%	42.45%	B-S	0.314	**3.13 ^†^**	S-O	0.746	1.50
Sex (S)	1	4.5%	0.00%	13.25%	B-G	0.247	1.80	S-M	0.470	**2.41 ^†^**
Gaming (G)	19	24.7%	15.05%	34.30%	B-O	0.290	**2.09 ^†^**	G-O	0.072	**3.75 ^†^**
Other (O)	3	13.6%	0.00%	27.98%	B-M	0.722	1.30	G-M	0.228	**2.33 ^†^**
Multiple (M)	3	14.3%	0.00%	29.25%	S-G	0.104	**5.63 ^†^**	O-M	0.605	1.61
Total	47	22.5%	16.83%	28.15%						
	**FA Symptom Severity (YAS-2 Total Score)**				**Pairwise Comparisons**					
	**Mean**	**SD**	**95% CI**		**Contr.**	**^2^ *p***	** *|d|* **	**Contr.**	**^2^ *p***	** *|d|* **
Buying (B)	3.14	3.86	2.25	4.03	B-S	**0.049 ***	**0.69 ^†^**	S-O	0.693	0.19
Sex (S)	1.43	2.03	0.04	2.81	B-G	0.315	0.26	S-M	0.363	0.40
Gaming (G)	2.49	3.02	1.73	3.24	B-O	0.099	**0.53 ^†^**	G-O	0.400	0.33
Other (O)	1.82	2.10	0.49	3.15	B-M	0.337	0.32	G-M	0.826	0.08
Multiple (M)	2.31	3.35	0.95	3.68	S-G	0.170	**0.52 ^†^**	O-M	0.614	0.22
Total	2.50	3.29	2.05	2.95						

Note. ^1^ Comparison between the groups based on logistic regression adjusted for sex. ^2^ Comparison between the groups based on ANCOVA adjusted for sex. FA+: Positive screening based on YFAS-2. Prev: prevalence. 95% CI: 95% confidence interval. Contr.: contrast. SD: standard deviation. * Bold: significant comparison (0.05 level). ^†^ Bold: relevant effect size (coefficient into the mild-moderate to large-high effect range).

**Table 3 nutrients-17-01279-t003:** Comparison of the clinical profile at baseline: ANCOVA adjusted for sex.

	FA−		FA+			
	(*n* = 162)		(*n* = 47)			
	Mean	SD	Mean	SD	*p*	*|d|*
Age of onset of BA (years)	27.37	14.25	23.56	11.16	0.115	0.30
Duration of the BA (years)	5.93	5.83	6.58	5.67	0.547	0.11
BMI (kg/m^2^)	25.24	4.96	28.51	6.76	**0.001 ***	**0.55 ^†^**
SCL-90R Somatization	1.05	0.79	1.70	1.10	**<0.001 ***	**0.68 ^†^**
SCL-90R Obsess.-compulsive	1.48	0.89	2.14	1.02	**<0.001 ***	**0.69 ^†^**
SCL-90R Interpersonal sensitivity	1.29	0.91	2.11	1.03	**<0.001 ***	**0.84 ^†^**
SCL-90R Depression	1.61	0.98	2.37	1.08	**<0.001 ***	**0.74 ^†^**
SCL-90R Anxiety	1.12	0.88	1.72	1.16	**<0.001 ***	**0.58 ^†^**
SCL-90R Hostility	0.95	0.76	1.65	1.15	**<0.001 ***	**0.71 ^†^**
SCL-90R Phobic anxiety	0.65	0.79	1.41	1.18	**<0.001 ***	**0.76 ^†^**
SCL-90R Paranoid ideation	1.10	0.84	1.73	0.99	**<0.001 ***	**0.69 ^†^**
SCL-90R Psychotic ideation	0.91	0.76	1.58	0.97	**<0.001 ***	**0.77 ^†^**
SCL-90R GSI	1.19	0.72	1.88	0.90	**<0.001 ***	**0.84 ^†^**
SCL-90R PST	49.03	20.96	62.44	19.34	**<0.001 ***	**0.67 ^†^**
SCL-90R PSDI	2.04	0.57	2.53	0.65	**<0.001 ***	**0.80 ^†^**
UPPS-P Lack of premeditation	24.10	6.61	24.70	7.69	0.617	0.08
UPPS-P Lack of perseverance	23.88	5.50	25.01	6.36	0.264	0.19
UPPS-P Sensation seeking	25.94	7.97	27.09	9.33	0.414	0.13
UPPS-P Positive urgency	28.49	9.51	33.91	10.98	**0.002 ***	**0.53 ^†^**
UPPS-P Negative urgency	30.41	6.81	35.34	7.59	**<0.001 ***	**0.68 ^†^**
UPPS-P Total score	132.82	20.55	146.04	27.10	**0.001 ***	**0.55 ^†^**
DERS Non-acceptance	15.76	4.94	20.32	5.68	**<0.001 ***	**0.86 ^†^**
DERS Goal-directed behaviors	15.48	3.01	18.88	4.12	**<0.001 ***	**0.94 ^†^**
DERS Difficulties in impulse control	13.11	3.77	17.87	6.33	**<0.001 ***	**0.91 ^†^**
DERS Lack of awareness	17.12	3.64	17.82	4.71	0.409	0.17
DERS Limited access to emotions	19.92	4.85	26.94	6.99	**<0.001 ***	**1.17 ^†^**
DERS Lack of emotional clarity	12.07	2.89	14.92	4.66	**<0.001 ***	**0.74 ^†^**
DERS Total score	93.95	14.85	116.72	25.40	**<0.001 ***	**1.09 ^†^**
TCI-R Novelty seeking	104.59	15.63	102.55	16.25	0.451	0.13
TCI-R Harm avoidance	105.87	18.20	122.12	18.66	**<0.001 ***	**0.88 ^†^**
TCI-R Reward dependence	96.04	16.59	94.13	16.93	0.502	0.11
TCI-R Persistence	98.74	20.80	90.52	22.63	**0.028 ***	0.38
TCI-R Self-directedness	126.57	19.87	112.43	24.96	**<0.001 ***	**0.63 ^†^**
TCI-R Cooperativeness	130.91	17.15	128.25	18.61	0.381	0.15
TCI-R Self-transcendence	61.39	15.19	62.62	14.90	0.649	0.08

Note. FA−: negative food addiction screening. FA+: positive food addiction screening. BMI: body mass index. SD: standard deviation. * Bold: significant comparison (0.05 level). ^†^ Bold: relevant effect size (coefficient into the mild-moderate to large-high effect range).

## Data Availability

The datasets presented in this article are not available due to the mandatory use of the protection law for anonymized data by public hospitals in Spain. Requests to access the datasets should be directed to sjimenez@bellvitgehospital.cat.

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
