# Peer review of "Exploring Food Addiction Across Several Behavioral Addictions: Analysis of Clinical Relevance"

_nutrients, 2025, doi:10.3390/nu17071279_

Round 1

Reviewer 1 Report

Comments and Suggestions for Authors

Dear Authors,
This study was conducted to exploring food addiction in several behavioral addictions. As you know, this topic already has very well-known world widely. Namely, it was so well-known topic that it is generalized and organized into textbooks. For this reason, it is difficult to confirm the originality, scientific nature, creativity, and novelty of this study.

Furthermore, checking by the iThenticate system, the plagiarism rate was 36% (quotes included and bibliography excluded). It is also serious flaws. You have to reduce the plagiarism rate under 10~15%.

Please refer to previous similarly studies.

Krupa H, Gearhardt AN, Lewandowski A, Avena NM. Food Addiction. Brain Sci. 2024 Sep 24;14(10):952. doi: 10.3390/brainsci14100952. PMID: 39451967; PMCID: PMC11506718.

LaFata EM, Allison KC, Audrain-McGovern J, Forman EM. Ultra-Processed Food Addiction: A Research Update. Curr Obes Rep. 2024 Jun;13(2):214-223. doi: 10.1007/s13679-024-00569-w. Epub 2024 May 18. PMID: 38760652; PMCID: PMC11150183.

di Giacomo E, Aliberti F, Pescatore F, Santorelli M, Pessina R, Placenti V, Colmegna F, Clerici M. Disentangling binge eating disorder and food addiction: a systematic review and meta-analysis. Eat Weight Disord. 2022 Aug;27(6):1963-1970. doi: 10.1007/s40519-021-01354-7. Epub 2022 Jan 18. PMID: 35041154; PMCID: PMC9287203.

Wiss DA, Avena N, Gold M. Food Addiction and Psychosocial Adversity: Biological Embedding, Contextual Factors, and Public Health Implications. Nutrients. 2020 Nov 16;12(11):3521. doi: 10.3390/nu12113521. PMID: 33207612; PMCID: PMC7698089.

Hauck C, Cook B, Ellrott T. Food addiction, eating addiction and eating disorders. Proc Nutr Soc. 2020 Feb;79(1):103-112. doi: 10.1017/S0029665119001162. Epub 2019 Nov 20. PMID: 31744566.

Adams RC, Sedgmond J, Maizey L, Chambers CD, Lawrence NS. Food Addiction: Implications for the Diagnosis and Treatment of Overeating. Nutrients. 2019 Sep 4;11(9):2086. doi: 10.3390/nu11092086. PMID: 31487791; PMCID: PMC6770567.

Petry NM, Zajac K, Ginley MK. Behavioral Addictions as Mental Disorders: To Be or Not To Be? Annu Rev Clin Psychol. 2018 May 7;14:399-423. doi: 10.1146/annurev-clinpsy-032816-045120. PMID: 29734827; PMCID: PMC5992581.

Gearhardt AN, Schulte EM. Is Food Addictive? A Review of the Science. Annu Rev Nutr. 2021 Oct 11;41:387-410. doi: 10.1146/annurev-nutr-110420-111710. Epub 2021 Jun 21. PMID: 34152831.

Jacques A, Chaaya N, Beecher K, Ali SA, Belmer A, Bartlett S. The impact of sugar consumption on stress driven, emotional and addictive behaviors. Neurosci Biobehav Rev. 2019 Aug;103:178-199. doi: 10.1016/j.neubiorev.2019.05.021. Epub 2019 May 21. PMID: 31125634.

Piccinni A, Bucchi R, Fini C, Vanelli F, Mauri M, Stallone T, Cavallo ED, Claudio C. Food addiction and psychiatric comorbidities: a review of current evidence. Eat Weight Disord. 2021 May;26(4):1049-1056. doi: 10.1007/s40519-020-01021-3. Epub 2020 Sep 23. PMID: 32968944.

Skinner J, Jebeile H, Burrows T. Food addiction and mental health in adolescents: a systematic review. Lancet Child Adolesc Health. 2021 Oct;5(10):751-766. doi: 10.1016/S2352-4642(21)00126-7. Epub 2021 Jun 24. PMID: 34174201.

Author Response

Thank you.

Reviewer 2 Report

Comments and Suggestions for Authors

Abstract
1.    The objectives can be more concisely structured. The phrase “also a secondary objective is to determine…” disrupts the flow and should be integrated more smoothly.
2.    Consider restructuring the objectives into a more readable format, avoiding redundancy.
3.    The abstract lacks details on how FA was assessed. What criteria or diagnostic tools were used?
4.    How were general psychopathology and personality traits measured? Mentioning key instruments would enhance clarity.
5.    A brief mention of the statistical methods used for analysis would improve rigor.
6.    The phrase “greater general psychopathology related to greater emotional distress” is somewhat unclear—does this mean FA is linked to more psychopathology and distress, or is distress mediating the relationship?
7.     “Determine the FA prevalence” should be revised to “determine FA prevalence” for grammatical accuracy.
8.    The authors conclude that FA and BA comorbidity is "relatively high," but this could be more specific. Is 22.49% considered high relative to prior studies?
9.    The claim that FA is a coping mechanism for emotional distress is plausible but should be phrased more cautiously unless causality is established.

Introduction
1.    The authors effectively highlight the shift from categorical to dimensional and transdiagnostic approaches in mental disorders, setting the stage for the study's rationale.
2.    The authors provide an extensive overview of the existing research on FA and its associations with different BAs, including compulsive buying-shopping disorder (CBSD), gambling disorder, and gaming disorder.
3.    The authors effectively identify gaps in the literature, particularly the lack of studies beyond gambling disorder, strengthening their research rationale.
4.    Some sentences are overly complex and could be streamlined for better readability. Example: “It has been postulated that these disorders would be located along a continuum, with impulsivity at one extreme and compulsivity at the other, however, this relation might be more complicated [4].”
5.    The sentence “This dimensional concept can group a series of symptoms that overlap and complement each other, forming a continuum in which different disorders may share similar characteristics [2]” should be more concise to facilitate understanding. 
6.    The transitions between topics do not always make clear connections. For instance, the discussion on transdiagnostic models and impulsive-compulsive spectrum disorders could better lead into FA’s relevance in this framework.
7.    The sentence: “Substance and non-substance-related disorders often coexist, with a comorbidity ranging from 57.5% to 76.3% [7–10].” can be better integrated by explicitly linking it to FA and behavioral addictions rather than presenting it as a standalone fact.
8.    The phrase “several studies have postulated common components in terms of etiology, course, and neurobiology” is later restated similarly. Consolidating these points would improve conciseness.
9.    The phrase “Recently, interest in studying FA in BA has increased” is somewhat redundant after establishing FA’s relevance in behavioral addictions.
10.    Several points need to be clarified. The authors state that “FA shares characteristics with gambling disorder [33] and other BAs, such as CBSD [36].” It would be beneficial to specify these shared characteristics (e.g., impulsivity, compulsivity, craving-related behaviours). Müller et al. [36] study on FA and BA in bariatric surgery patients is mentioned. Still, clarifying whether the findings are generalizable to non-bariatric populations would be helpful.
11.    The reference “[17–19] [20–22]” appears to have an extra set of brackets, which should be corrected.
12.    The phrase “d), also a secondary objective, is to determine whether the presence of FA is linked to a higher mean body mass index (BMI).” It should be restructured for better clarity. 
13.    The statement “To our knowledge, this is the first clinical population study to investigate FA in treatment-seeking individuals with gaming disorder, CBSD, and compulsive sexual behavior disorder as primary diagnoses.” is a strong claim. Still, briefly clarifying how this study differs from existing literature would be helpful. For example, is it the first to use a specific methodology or examine all three disorders together?
14.    The authors state: “We hypothesize that participants with FA and these BAs will have a specific profile that differentiates them from other patients who do not have FA.” It would strengthen the hypothesis if they explicitly referred to supporting literature or prior findings indicating that FA co-occurs with more severe psychopathology.

Methods
1.    The description of participants, including sample size, gender distribution, and recruitment site, is clear and specific.
2.    The exclusion of participants with eating disorders (EDs) is well justified to avoid confounding effects.
3.    A comprehensive set of validated instruments is used to assess behavioral addictions (BAs), psychopathology, personality traits, impulsivity, and emotion regulation.
4.    Appropriate statistical tests (Chi-square, ANOVA, logistic regression, ANCOVA) are employed to compare groups.
5.    The application of Finner’s correction helps control for Type I error inflation, improving the robustness of findings.
6.    It is unclear how patients were approached and screened for participation. Were all patients at the Behavioral Addictions Unit invited, or was there a pre-screening?
7.    How many individuals were initially assessed, and how many were excluded based on the eligibility criteria? Reporting dropout rates would enhance transparency.
8.    The compulsive sexual behaviour disorder diagnosis relies on a self-reported item list rather than structured clinical interviews, which may introduce bias. Consider validating self-reported diagnoses against a clinical interview or at least mention this.
9.    While excluding eating disorders is stated, clarifying whether this was done to control for confounding variables or due to the study’s objectives would be helpful. If eating disorders have a known relationship with behavioral addictions, their exclusion should be explicitly justified.
10.    Given that the DSM-IV-TR is outdated, why were these criteria used instead of more recent conceptualizations? Consider discussing the rationale for using older classification criteria.
11.    Were assumptions for logistic regression and ANCOVA tested (e.g., normality, homoscedasticity, multicollinearity)? A brief mention of assumption checks would strengthen the validity of the results.
12.    Were there any missing data points, and how were they handled (e.g., multiple imputation, listwise deletion)? Addressing this would improve methodological transparency.
Results
1.    The demographic characteristics (gender, marital status, education level, employment status, social position index) are well detailed.
2.    The study effectively presents FA-positive screening rates across different behavioral addiction (BA) subtypes.
3.    The use of ANCOVA to adjust for gender in comparisons is methodologically sound.
4.    How was the FA-positive screening threshold determined? Was it based on prior literature or empirical validation?
5.    Consider citing references or providing a rationale for the cutoff point in the YFAS-2.
6.    The authors control for gender in comparisons, but are there other potential confounders (e.g., age, socioeconomic status, comorbid mental health conditions) that might influence FA prevalence?
7.    Adjusting for additional covariates could improve the robustness of the results.
8.    The FA severity measure is reported as a mean (SD=3.3), but was normality assessed before applying ANCOVA? Consider reporting median values or using non-parametric alternatives if the distribution was skewed.
9.    The authors state that Table 3 displays ANCOVA results, but it would be helpful to summarize key findings explicitly.
10.    Were any post-hoc tests conducted to explore significant differences between groups?
11.    While statistical significance is implied, reporting effect sizes (Cohen’s d, eta squared, or odds ratios for categorical comparisons) would provide a better understanding of clinical relevance.

Discussion
1.    The discussion shifts between FA as a transdiagnostic construct, a comorbidity, and a distinct condition. Clarifying this conceptual framework would strengthen coherence.
2.    The term "FA+BA" is used frequently but is not clearly defined. Does it refer to all BAs combined or to specific BA subtypes?
3.    The finding that FA+BA comorbidity is more common in females aligns with previous literature, but causality is not addressed.
4.    Are there potential psychosocial or neurobiological explanations for this trend? Consider elaborating on mechanisms (e.g., hormonal influences, coping styles).
5.    The authors note that emotional impulsivity is heightened in FA+BA individuals but do not explore why cognitive impulsivity remains unaffected.
6.    The role of reinforcement mechanisms (positive vs. negative) in FA and BA interactions can be expanded.
7.    The higher BMI in FA+BA individuals is acknowledged, but potential confounding factors (e.g., metabolic differences, lifestyle factors, medication effects) are not discussed.
8.    The statement that FA can occur across weight categories is essential but needs further elaboration.
9.    The recommendation to incorporate FA screening in BA treatment is strong, but how should this be implemented in clinical settings?
10.    Mentioning "transdiagnostic treatments" is promising. Can specific therapeutic approaches (e.g., DBT, ACT, and personalized nutrition interventions) be suggested?
11.    The discussion and limitations already cover some points in the conclusion (e.g., FA as a severe and dysfunctional comorbidity, the need for more research). Streamlining this section by focusing on the most novel contributions would improve readability.

Comments on the Quality of English Language

English can be improved. 

Author Response

Thank you.

Reviewer 3 Report

Comments and Suggestions for Authors

The article Exploring Food Addiction in several Behavioral Addictions: Analysis of Clinical Relevance deals with the critical and current problem of the co-occurrence of food addiction (FA) with other behavioral addictions (BA). The study aimed to determine the frequency of FA among patients treated for disorders such as gaming disorder, compulsive buying disorder (CBSD), and compulsive sexual behavior, as well as to analyze their psychopathological and sociodemographic characteristics.

Strengths of the study

The study makes an essential contribution to research on the comorbidity of FA and BA, filling a gap in the literature, which has so far focused mainly on gambling. The authors use a transdiagnostic approach, which is in line with the contemporary model of mental disorders and takes into account the spectrum of impulsivity and compulsivity symptoms. The value of the work is the use of a multidimensional assessment of psychopathology and the use of verified research tools, such as the Yale Food Addiction Scale 2.0, the Symptom Checklist-Revised (SCL-90-R), and the UPPS-P Impulsive Behavior Scale, which increases the reliability of the results. The statistical analysis is carried out correctly and adequately adjusted for confounding variables such as gender.

Limitations of the study

The main limitation is the nature of the sample – patients were recruited from only one clinical center, which limits the possibility of generalizing the results. There is no control group from the general population, which makes it impossible to determine whether the differences in the frequency of FA are due to the presence of BA or other factors. Furthermore, measurements such as BMI are based on patients' self-reported data, which can lead to errors. The results are based on a cross-sectional study model, which makes it impossible to determine the direction of the cause-and-effect relationship - whether FA leads to BA or vice versa. The influence of genetic or neurobiological factors, which could be crucial for understanding the mechanisms of this comorbidity, has also not been considered.

Recommendations for improving the article

1. The sample selection procedure and its representativeness for the clinical population must be described in more detail, as well as the diagnostic tools, especially regarding the FA criteria.

2. I suggest introducing an additional regression analysis to determine which factors are independent predictors of FA among people with BA. It is currently unclear whether impulsivity, depressive symptoms, or BMI are the cause or effect of FA.

3. The conclusions are consistent with the results but require greater precision.

It is worth stating more clearly whether FA should be treated as an independent disorder or a co-occurring BA feature. The current wording suggests that FA is a phenomenon of great clinical significance, but the article does not provide sufficient evidence to recognize it as a separate diagnostic entity.

4. It is worth considering the perspective of longitudinal and neurobiological research and verifying the effectiveness of treating people with FA+BA compared to people with isolated BA.

5. The article lacks references to studies on therapeutic interventions in FA and neuroimaging studies that could confirm the hypothesis of common neuronal mechanisms of FA and BA. It is worth supplementing the literature with meta-analyses on the effectiveness of FA and BA treatment.

Summary

The article deals with an important topic and adds value to the literature, but it requires significant revisions, especially regarding methodology and interpretation of results. Without these changes, the publication may be premature because the current data do not allow for an unambiguous determination of the relationship between FA and BA. Formulating the conclusions more precisely and relating them to clinical practice is also necessary. For this reason, I recommend major revisions before acceptance for publication.

Author Response

Thank you.

Round 2

Reviewer 1 Report

Comments and Suggestions for Authors

Again. I believe this study’s findings and interpretations are rudimentary (interpretation of the results are beginner level) and lack sufficient detail to be valuable to the science. The Nutrients journal is Q1 journal (IF=4.8), therefore, the quality of this manuscript is not good enough to be published in this journal.

Moreover, authors mentioned that “Regarding the issue of plagiarism, we have found that various repetitions concerning the authors, affiliations, acknowledgments and funding sections, which cannot be modified.” However, although all manuscripts are same situation about plagiarism issue, it was controlled under 15 % plagiarism.

This manuscript is still the plagiarism rate was 32% (quotes included and bibliography excluded). It is really serious flaws and problems.

Author Response

We thank the reviewer for taking the time to assess our manuscript once more. It is very important for us to present a quality piece of work, consequently we are executing all the necessary changes to do so. Undoubtedly, the reviewer's comments will help us to achieve this goal. Therefore, if possible, could the reviewer tell us in which sections we may improve the manuscript, or any changes that could be made to specific sections, it would be greatly appreciated.

Regarding the issue of plagiarism, we thank the reviewer for their feedback and share their concern in this regard. We have requested a subsequent review of the plagiarism percentage and with this information, we have modified the wording in various portions of the manuscript. We have submitted an updated version of it. However, we have to emphasize, that we have once again verified that this percentage includes portions which cannot be altered, i.e., affiliations, glossary, portions of the description of the psychometric procedures, referencing specific statistical analyses used, acknowledgments and funding sources. Additionally, due to the importance of this issue, we have also made sure that none of the sentences which have been marked as being plagiarized have to do with uncited work from previous studies (i.e., in the introduction or discussion sections) and that the intended authors have been credited.

Any changes or recommendations which could help us to improve the quality of this work are greatly valued.

Thank you very much for your time.

Reviewer 3 Report

Comments and Suggestions for Authors

The authors have addressed all comments in full and made appropriate changes to the manuscript. They have taken into account both methodological and interpretative issues and have made the necessary additions regarding the representativeness of the sample, statistical analysis, the conceptual positioning of FA, references to neurobiological research, and the effectiveness of treatment. Therefore, I accept the revised version and thank the authors for kindly considering my comments.

Author Response

We thank the reviewer for carefully reading our manuscript and for their valuable suggestions to improve the quality of our work. We thank you for your kindness, your time and the opportunity to learn from you.